# SYT1-Associated Neurodevelopmental Disorder: A Narrative Review

**DOI:** 10.3390/children9101439

**Published:** 2022-09-22

**Authors:** Edith Riggs, Zaynab Shakkour, Christopher L. Anderson, Paul R. Carney

**Affiliations:** 1College of Osteopathic Medicine, Kansas City University School of Medicine, Kansas City, MO 64106, USA; 2School of Medicine, University of Missouri Child Health, Columbia, MO 65201, USA; 3Department of Engineering, University of Missouri Biomedical Engineering, Columbia, MO 65201, USA

**Keywords:** intellectual disability, synaptic vesicles, synapse, neurodevelopment, synaptotagmin, genetic disorder, neurotransmission

## Abstract

Synaptic dysregulations often result in damaging effects on the central nervous system, resulting in a wide range of brain and neurodevelopment disorders that are caused by mutations disrupting synaptic proteins. SYT1, an identified synaptotagmin protein, plays an essential role in mediating the release of calcium-triggered neurotransmitters (NT) involved in regular synaptic vesicle exocytosis. Considering the significant role of SYT1 in the physiology of synaptic neurotransmission, dysfunction and degeneration of this protein can result in a severe neurological impairment. Genetic variants lead to a newly discovered rare disorder, known as SYT1-associated neurodevelopment disorder. In this review, we will discuss in depth the function of SYT1 in synapse and the underlying molecular mechanisms. We will highlight the genetic basis of SYT1-associated neurodevelopmental disorder along with known phenotypes, with possible interventions and direction of research.

## 1. Introduction

Synaptotagmins (SYTs) are an evolutionary conserved family of membrane-trafficking proteins composed of an N-terminal transmembrane domain (TMD), a variable linker, and two cytoplasmic C2 domains (C2A and C2B domains) at the C-terminus of the protein [1]. The mammalian synaptotagmin family includes 16 genes [2] and 17 isoforms [3], most of which are exclusively expressed in the nervous system. Nevertheless, the level and location of expression throughout the brain is different between different isoforms [4], as well as their subcellular localization and function [5]. Among the identified isoforms, SYT1 remains the most extensively studied. Genetic and functional studies revealed that SYT1 has an essential role in mediating the release of calcium-triggered neurotransmitters (NT) involved in regular synaptic vesicle exocytosis [2,6]. Calcium binds via tandem C2-domains, C2A and C2B, of SYT1 thereby facilitating synaptic membrane fusion with the presynaptic membrane [7]. Additionally, SYT1 plays a regulatory role in endocytosis, functioning as a crucial vesicle cargo molecule [8].

Considering the significant role of SYT1 in the physiology of synaptic neurotransmission, it has been investigated as a potential biomarker for synaptic dysfunction and degeneration in several neurological disorders [9,10,11]. Synaptic dysregulations have deleterious effects on the central nervous system. Through genomic and proteomic sequencing of synapse, it has been shown that a wide range of brain disorders are caused by mutations disrupting synaptic proteins [12,13]. For instance, autism spectrum disorder (ASD), a neurodevelopmental disorder characterized by impaired social interactions and stereotypic behaviors [14], has been associated with genetic mutations in post-synaptic adhesion molecules [15,16]. Other complex psychiatric and neurological disorders, including schizophrenia (SCZ) [17], major depressive disorder (MDD) [18], and Alzheimer’s disease (AD) [19], were shown to be linked to dysregulations in synaptic transmission. Clinical features range from mild to severe and include abnormal involuntary movements, motor deficits, behavioral abnormalities, and developmental delay [9,20,21]. In this review, we will discuss in depth the function of SYT1 in synapse and the underlying molecular mechanism. Additionally, we will highlight the genetic basis of SYT1-associated neurodevelopmental disorder along with possible interventions.

## 2. Methodology

To perform this review, the authors performed searches related to the topic through PubMed using a combination and mixture of keyword-based and controlled vocabulary search strategies. Key concepts were identified and combined with Medical Subject Headings (MeSH) key terms as follows:

((((((((((((((((((“Synaptotagmins”[Mesh] AND “Synaptotagmin I”[Mesh]) AND “Vesicular Transport Proteins”[Mesh]) AND “SNARE Proteins”[Mesh]) OR “Protein Isoforms”[Mesh]) AND “Synaptic Transmission”[Mesh]) AND “Excitatory Postsynaptic Potentials”[Mesh]) AND “Inhibitory Postsynaptic Potentials”[Mesh]) AND “Postsynaptic Potential Summation”[Mesh]) AND “Receptors, Neurotransmitter”[Mesh]) AND “Child Behavior Disorders”[Mesh]) AND “Neurodevelopmental Disorders”[Mesh]) AND “Intellectual Disability”[Mesh]) AND “Child Development Disorders, Pervasive”[Mesh]) AND “Proteins”[Mesh]) OR “Phenotype”[Mesh]) AND “Genetic Markers”[Mesh]) OR “Vesicular Neurotransmitter Transport Proteins”[Mesh]) AND “Syt1 protein, rat”).

## 3. Synaptotagmin Isoforms and SYT1 Function

Synaptic neurotransmission is the fundamental communication between neurons. At the level of synapse, a transmitter is released from one neuron generating a signal on another neuron or target cell that can excites, inhibits, or modulates its cellular activity [22]. The action potential reaching the nerve terminal initiates a rapid influx of Ca^2+^ions into the postsynaptic cell, causing fusion of vesicles and ultimately the release of neurotransmitters from those vesicles (Figure 1). This release is mediated via the assembly and zippering of SNARE (soluble N-ethylmaleimide-sensitive fusion protein (NSF) attachment protein receptor) complexes that allow the destined membranes to fuse [23]. SNARE complexes in neurons have three components, synaptobrevin or vesicle-associated membrane protein (VAMP), which is the v-SNARE, and two plasma-membrane proteins syntaxin and SNAP-25 (synaptosome-associated protein, 25 kDA) that are the t-SNARES. Once these two membranes combine, they form four-helix bundle aligned in a parallel fashion called the trans-SNARE complex [24]. This machinery is essential for vesicle docking and priming at active zones. Previous studies have indicated the role of SYT1 as a dual Ca^2+^sensor able to carry out both endocytosis and exocytosis independently of one another [1]. These processes are carried out by SYT1’s C2A and C2B domain in a Ca^2+^dependent manner.

Synaptotagmins and SNARE proteins play a pivotal role in evoked synchronous neurotransmitter release in a Ca^2+^ dependent and independent manner. For the purposes of this review, we will be discussing the SYT isoforms of clinical and functional relevance related to SYT1-associated neurodevelopmental disorder (Table 1). These isoforms include SYT2, SYT7, and SYT9 [25,26].

In the forebrain specifically, SYT2 seems to be concentrated in inhibitory neurons, whereas SYT1 dominates excitatory neurons [27]. Interestingly, it has been discovered that KO of SYT2 impairs Ca^2+^ induced vesicle release and produces severe motor deficits in adolescent mice [27]. In clinical studies, mutations in SYT2 have been reported in the C2B binding domain, similar to SYT1-associated neurodevelopmental disorder, resulting in motor deficits including muscle weakness, bulbar deficits, and delay in motor developmental that ranges from mild to moderate to severe depending on the mutation [28].

SYT9 has the slowest NT release kinetics and lowest affinity for Ca^2+^ [25,29]. It is concentrated in dense core vesicles and synaptic vesicles within the limbic system and striatum [27,29]. SYT9 is the major isoform responsible for fast-acting synaptic vesicle exocytosis in these neurons [27,29]. In addition to striatal neurons, SYT9 has also been found in cortical and hippocampal neurons It has been demonstrated in previous labs that all three of these neuron classes are regulated by SYT1 and SYT9 but have shown that loss of SYT1 inhibited evoked rapid NT release while the loss of SYT9 had no discernable effect [25,30]. Interestingly, it has been shown that SYT9 can rescue the loss of SYT1 but only when immensely overexpressed [25,27].

Unlike SYT1, C2A is the main Ca^2+^sensing domain in SYT9 [5,25] and mutations within this domain disrupts the ability for SYT9 to initiate membrane fusion in the presence of AP-driven Ca^2+^influx [25].

It has been found that SYT7 has redundant function with SYT1 and is capable of SV priming and clamping. It is a high affinity Ca^2+^sensor for slow asynchronous neurotransmitter release [26,30] and takes part in Ca^2+^dependent short term synaptic facilitation, SV endocytosis and is responsible for maintaining the RRP of vesicles [30,31]. Simultaneous loss of function of SYT1 and SYT7 demonstrated a decrease in RRP size in both excitatory and inhibitory synapses, whereas loss of function in only one or the other did not show any effect on RRP size [30]. Loss of function of both SYT1 and SYT7 does not seem to alter the rate of vesicle priming into RRP [30]. Due to redundancy in function, phenotypic loss of SYT1 can be partially compensated by SYT7, albeit in immature neurons [26].

SYT1 is responsible for Ca^2+^dependent and independent fast synchronous NT release [32] and plays a role in clamping mini spontaneous release and in the maintenance of readily releasable pools (RRP) in vesicles. SYT1 is the main isoform found to reside within synaptic vesicles and secretory granules in neurons and neuroendocrine cells located mainly in the hippocampus and forebrain [29]. SYT1 is a low affinity Ca^2+^sensor that binds to Ca^2+^following an action potential via the C2B domain initiating the fusion of the SNARE complex thereby providing the main mechanism for spontaneous and synchronous release [26,32,33]. C2B domain is the main site of mutations associated with SYT1-associated neurodevelopmental disorder. Such mutation has demonstrated decreased evoked synchronous release of NTs due to the inhibition of SV fusion [1,9,32]. The precise mechanism will be discussed.

In exocytosis, the SYT1 protein senses the rise in intracellular Ca^2+^due to an action potential and in turn interacts with negatively charged phospholipids and SNARE proteins to promote membrane fusion [25]. The interaction of SYT1 with negatively charged phospholipid phosphatidylserine is crucial to exocytosis and has been highlighted by previous experiments in C2A mutations that abolished the interaction between SYT1 and phospholipids [34]. However, this mutation in C2A has not had the same effects on the interaction between SYT1 and SNARE proteins [34]. Upon Ca^2+^binding, SYT1 acts by deforming the plasma membrane or binding to the curved membrane via the C2B domain [32]. This interaction is found to be critical in NT release and exocytosis [32]. Furthermore, when testing the function of SYT1 during exocytosis it was found that C2B mutations greatly disrupted this process whereas corresponding mutations in the C2A domain did not [1].

The efficiency of endocytosis and exocytosis is crucial to establishing efficient neuronal transmission. SYT1 helps to establish this via a bidirectional Ca^2+^ sensing capability that links endocytosis and exocytosis [35]. Previous labs have recognized that SYT1 is able to initiate slow and small clathrin mediated endocytosis (CME) and/or block bulk-sized endocytosis during increased neural activity [35]. In contrast to the pivotal role of C2B in exocytosis, it has been demonstrated that either C2B or C2A can be involved in SYT1-mediated endocytosis [1]. Interestingly, endocytosis can proceed in both a Ca^2+^ dependent and independent manner [1]. However, Ca^2+^binding to either the C2A or C2B domain is necessary in the acceleration of endocytosis [1]. Experiments have shown that mutations in either C2A or C2B (but not in both simultaneously) rescued knockdown-decrease in transferrin uptake, which represented CME and further established the redundancy of C2AB in endocytosis [35]. It was further demonstrated that C2B is needed for bulk endocytosis [35].

### 3.1. Mutations in C2AB Inhibit Exocytosis

Previous studies indicate that the most consequential clinical manifestations occur with mutations within the amino acid sequence bind domain of the C2B domain [1,9,20]. The C2B and C2A domain are structurally similar but the C2B domain has a higher affinity for Ca^2+^-dependent membrane penetration due to the positive cooperativity between C2B and PIP2 [36]. This positive cooperativity promotes the oligomerization of SYT1 [36]. Interestingly, recent studies have shown that both C2B and C2A work in a cooperative manner to assist SYT1 membrane binding. However, C2B still remains the dominant energetic domain in these transmission interactions [34,36,37].

The C2B domain is composed of a series of polylysine residues region at its side that bind to PIP2 in the absence of Ca^2+^ [38] and two basic residues region located at its bottom; both of which have been shown to be crucial in synchronous NT release [33]. The C2B domain forms a primary and secondary interface with the SNARE complex, which is critical in NT release and exocytosis [32]. A primary interface occurs between the C2B domain and SNARE complex via the interaction between arginine amino acids with glutamates and aspartates between SNAP25 and Syyntaxin-1 [25]. It has been demonstrated that mutations from arginine to glutamines in C2B show decreases in the rate of fusion between C2B and the SNARE complex, thus disrupting exocytosis [32].

Studies have demonstrated that the mutations in C2B domain did not affect the binding rate of Ca^2+^ to C2B, but rather the time bound decreased [9] and inhibition of NT release via a dominant-negative manner [26,36]. Mutations that neutralized the C2B domain resulted in completely abolishing evoked NT release and disrupt Ca^2+^ independent vesical docking [33,37]. However, a mutation in C2A resulted in only a 50% decrease in NT release [37]. It is thought that C2B might be the governing domain in evoked NT release due to its increased Ca^2+^ affinity and stronger membrane binding energy [37]. The mediating role of C2A is not completely understood on a molecular level and must be studied further.

### 3.2. Mutations in SYT1 Result in Synaptic Neuronal Transmission Dysfunction

It has been described by Baker et al. [20] that SYT1 mutation causes a dysfunction in synaptic vesicle cycling. In the years since, multiple missense mutations have been implicated in what has been classified as SYT1 -associated neurodevelopmental disorder. Each of the missense mutations causing SYT1associated neurodevelopmental disorder have been implicated in the C2B domain of SYT1 [1,9,20] thereby disrupting SYT1 function. Specifically, it has been demonstrated that mutations reduce the binding between C2B and PIP2 and significantly decrease the membrane sensitivity to Ca^2+^influx after an AP [38]. On a biomolecular level, this occurs when one of the hydrophobic amino acid residues within the C2B domain are replaced by a neutral or acidic amino acid residue that abolishes Ca^2+^ and phospholipid binding, which ultimately negatively effects NT release [9,20].

The consequences of these mutations in SYT1 include disrupted vesicle trafficking, mis-localization of SYT1, deficits in evoked synaptic transmission, dose-dependent dominant negative activity, and diminished rate of exocytosis and SV fusion in mammalian synaptic terminals [9,20,25]. However, the severity of the dysfunction depends on the specific mutation. For example, it has been described that while most mutations do impact spontaneous release rates and evoked synchronous NT release, the mutations D303G and I367T in C2B have been witnessed to have a severe impact on evoked synaptic release in comparison to D365E [25]. It has also been observed that the three before-mentioned mutations form unstable complexes with Ca^2+^, but, interestingly, have increased rapid release of Ca^2+^, clearly showing that these pathogenic mutations have a reduced Ca^2+^ sensitivity [25].

In addition, the first mutation identified in a neurodevelopmental disorder, I368T, was shown to decrease rate of SV NT release and rate of evoked SV fusion due to the isoleucine being replaced by threonine. Isoleucine is of particular interest considering that it is a highly conserved residue in the C2B domain that mediates Ca^2+^ binding [9,20]. Interestingly, previous labs have demonstrated that SYT1 variants had roughly the same level of expression at the nerve terminal as wild type SYT1 except for M303K, which showed decreased expression and more diffuse localization [9].

## 4. Genotype-Phenotypes Relationship in SYT1 Associated Neurodevelopmental Disorder

Since SYT1 Associated Neurodevelopmental Disorder was first described by the Baker and Gordon Lab in 2015 [20], fifteen variants have been discovered. The common variants are mostly missense mutations in the high affinity Ca^2+^ C2B domain of the SYT1 protein. These variants have been associated with severe developmental and cognitive delay, deficits in movement, and behavioral abnormalities [39]. Baker et al. has also described four mutations that occur in the C2A domain [39]. Although the functions of the areas of mutation within the C2A domain are not fully understood, instability within the domain has been observed [39].

What is most intriguing about the genotypic variants is the spectrum of observed phenotypes. Although there seem to be hallmark clinical presentations that occur across all variants, there is a spectrum ranging from mild to severe between variants with some symptoms occurring exclusively to specific mutations [9,39]. In instances where there have been multiple patients with the same mutation, there have been correlating characteristic phenotypes that are specific to these mutations [39]. Despite these early findings, more research must be conducted to ascertain if individual genotypes do in fact have similar phenotypes among patients. Knowing the severity of clinical symptoms based on genotype may assist clinicians to treat patients more effectively and in a targeted manner.

The severity of clinical symptoms may vary depending on the genotypic SYT1 variant, there are hallmark symptoms that are universal among patients. These include developmental delay, sleep disturbances, EEG abnormalities, abnormal motor function, and abnormal eye physiology [9,39]. Other symptoms that occur on a spectrum include mood and behavioral disturbances, seizures, delayed speech and motor function, and involuntary movements [9,20,39]. In the paper released by Baker et al. in 2021, there have been a total of fifteen genotypes identified so far, with most phenotypes described.

The first variant discovered that prompted the classification of SYT1-associated neurodevelopmental disorder was I368T in 2015. Movement abnormalities including hyperkinesis, chorea, and dystonia were observed with involuntary movements occurring in one third of the cohort [9,20]. Abnormal eye physiology including nystagmus, esotropia, and bilateral hypermetropia were identified. In addition, EEG abnormalities were seen similar to other variants: EEG readings displayed slow and high amplitude oscillations [20]. Self-harm and behavioral disturbances were also universal across patients with the I368T variant such head-banging, finger and nail biting and chest beating when agitated [9,20]. Seizures were not observed across all patients for this variant [9,20]. Interestingly, of the patients studied, there were other comorbidities described seemingly unrelated to the universal symptoms seen. These included gastro-esophageal reflux and hyperventilation-induced cyanotic episodes. However, there has been no explanation thus far identifying the relationship between these other illnesses and SYT1-associated neurodevelopmental disorder [9,20].

M303K is another variant within the C2B domain that displays a phenotypic range of neurological and neurodevelopmental symptoms similar to the other variants. Movement abnormalities were noted and limited to ataxia and impaired fine motor skills [9]. However, in the one patient observed, no other involuntary movements were identified. Motor development, speech and language development were mildly [9]. When comparing intellectual disability across the C2B domain variants, it was reported that the individual with the M303K variant had a more largely utilized vocabulary and thus a less severe intellectual delay [9]. No seizures were reported [9]. Ophthalmic abnormalities were limited to esotropia [9]. Behavioral disturbances including angry outbursts and impulsivity were reported but no instances of self-harm [9]. More patients with M303K must be observed in order to establish any commonality among clinical presentations.

One patient’s phenotype was reported with the variant D304G. This patient did seem to have milder symptoms throughout but did have significant movement abnormalities including progressive lower extremity contractures and scoliosis [9]. It is necessary to record the extent of movement disorders in other patients with D304K. This patient also developed severe gastroesophageal reflux [9]. Ophthalmic abnormalities were noted including strabismus and hypermetropia [9]. Behavioral disturbances and self-harm including head banging, chest beating, and mouthing objects were also reported [9].

The phenotype of D366E were observed in three patients. Ophthalmic abnormalities were found including nystagmus, esotropia, and strabismus [9]. These patients did not present with involuntary movements or motor impairments were found in this small cohort, but they did exhibit self-harming behaviors including biting, scratching, hand-biting, chest banging, and emotional disturbances such as screaming and obsessions/repetitions [9]. Interestingly, two of the three patients presented with unique comorbidities that the patients with other variants did not have. These include atrial septal defect, laryngomalacia, syndactyly, lumbar lordosis, and bilateral hindfoot valgus deformities [9]. While there is no mention of remarkable medical histories for these three patients in particular, more investigation is needed to ascertain if these comorbidities are a downstream effect of SYT1 pathogenesis or a consequence of some other mechanism.

Two patients with the mutation N371K were identified. Each displayed extensive post-infantile dystonia and involuntary movement abnormalities including dyskinetic cerebral palsy, trunk and limb dystonia, and chorea [9]. Nystagmus was observed in both patients. Both patients exhibited forms of self-harm including screaming episodes, teeth-grinding, and hand chewing. The two patients had additional comorbidities including feeding difficulties, sleep apnea, and gastroesophageal reflux. What is most important to note within this cohort is that in all patients presented with infantile hypotonia but did not have seizures [9]. Six other C2B variants have been identified in Baker et al.’s most recent research published in 2021 [39]. However, phenotypes for these additional variants have yet to be established and thus require more research. This statement extends to the four C2A variants found.

## 5. Interventions and Promising Pharmaceuticals

There is no cure or definitive pharmaceutical medicine that can directly treat SYT1. Instead, a multifactorial approach has been taken by clinicians in order to alleviate adverse clinical symptoms that arise, such as sleep deficits, motor deficits, behavioral issues, and others.

It was reported by Baker et al. those various patients suffering from sleep disturbances and hyperventilation-induce cyanotic episodes were treated with clonidine to reduce these symptoms [9]. Another patient with a missense mutation I368T was treated with dopamine agonist pramipexole, which has been used successfully to treat mood disorders such as bipolar disorder, sleep disorders, Parkinson’s disease, and restless leg syndrome [9]. However, further investigation is required to assess the efficacy and effectiveness of drugs such as pramipexole and its role in improving the clinical symptoms mentioned previously. While it was originally reported that patients with SYT1 Associated Neurodevelopmental Disorder did not suffer from seizures, it has been more recently debated that they, in fact, do suffer from seizures and it is widely accepted that a universal hallmark of SYT1-associated neurodevelopmental disorder is an abnormal EEG [39]. It has been clinical practice to treat these patients with various anti-epileptic medications, but further investigation is needed to delineate the type, extent, and severity of seizures that the patients experience as well as the outcomes of anti-epileptic pharmaceutical treatment.

It was described by Baker et al. in 2018 that deficits in synaptic transmission created by SYT1 variants was rescued only when there was a significant increase in extracellular Ca^2+^ levels [9]. Although this has yet to be achieved in a clinical setting with patients, there has been experimentation with 4-Aminopyridine and has seen some success. 4-Aminopyridine is a K+ channel antagonist that has been clinically approved to treat multiple sclerosis [25,40]. 4-Aminopyridine has been found to stimulate NT exocytosis, mainly glutamate, from nerve terminals both in an Ca^2+^ independent and dependent manner [40]. In the absence of external Ca^2+^, 4-Aminopyridine was able to evoke NT release from synaptic vesicles even below the amount needed for membrane depolarization [40]. It was observed in another study that 4-Aminopyridine was able to increase glutamate release from the nerve terminal and restore short term plasticity when administered dose-dependently to neurons in vitro [25]. Although this shows potential to address impaired exocytosis in SYT1 patients, it is necessary to study this further in animal models and assess the extent to which synaptic transmission is restored.

## 6. Discussion and Future Work

Currently, there are fifteen known missense mutations located in the C2B and C2A domains, with C2B being more common. Increased genetic testing may lead to discovery of additional mutations within these domains or elsewhere within the SYT1 protein that severely alters synaptic transmission. More research is required to investigate this as well as to assess the extent of clinical manifestations. What is most interesting about SYT1 Associated Neurodevelopmental Disorder is the variability of phenotypic symptoms in relation to genotype. So far, research groups have been able to compare phenotypes from one variant to another as well between patients who have the same mutation. However, this work is limited, and more research is needed to identify common and outlying biomarkers. If this were to be achieved, it may assist scientists to effectively target and treat those symptoms. 

This concept extends to the mechanism of action of various genotypic variants. While some have been well described over the years, the mechanism of action of other variants is largely unknown. We must first establish the biomolecular pathogenesis and extent of clinical symptoms for those variants in order to begin investigating potential productive therapeutic methods of treatment.

Unfortunately, there is no therapy that currently targets SYT1 and alleviates impaired synchronous NT exocytosis. While there have been some drugs experimented with, most are used to lessen the severity of associated symptoms. This is the reality of treating patients with SYT1 Associated Neurodevelopmental Disorder. The next logical step to creating an effective treatment targeting SYT1 specifically would be to develop an animal model and identify any potential modifiers or amplifiers for each of the variants. Additionally, further Natural History Studies focusing on expanding the known patient population and establishing the severity in phenotypes correlating with the specific SYT1 missense. When this is accomplished, a better understanding of potential treatment options can be considered and pharmaceuticals can be tested or developed to target said modifiers/amplifiers.

## 7. Conclusions

SYT1 associated neurodevelopmental disorder is a recently discovered disease with little known about the mechanism of action, the extent of the mutations within the SYT1 protein, and the severity of clinical manifestations. It was reported in 2015 that eleven patients were diagnosed with SYT1-associated neurodevelopmental disorder, but it is likely that, with significant improvements and availability of clinical genetic testing, a significantly larger population of SYT1 mutations will be found. Increased availability of precise genetic testing has made it possible to effectively diagnose patients and may prove that SYT1 Associated Neurodevelopmental Disorder is more common than we had previously assumed.

## Figures and Tables

**Figure 1 children-09-01439-f001:**
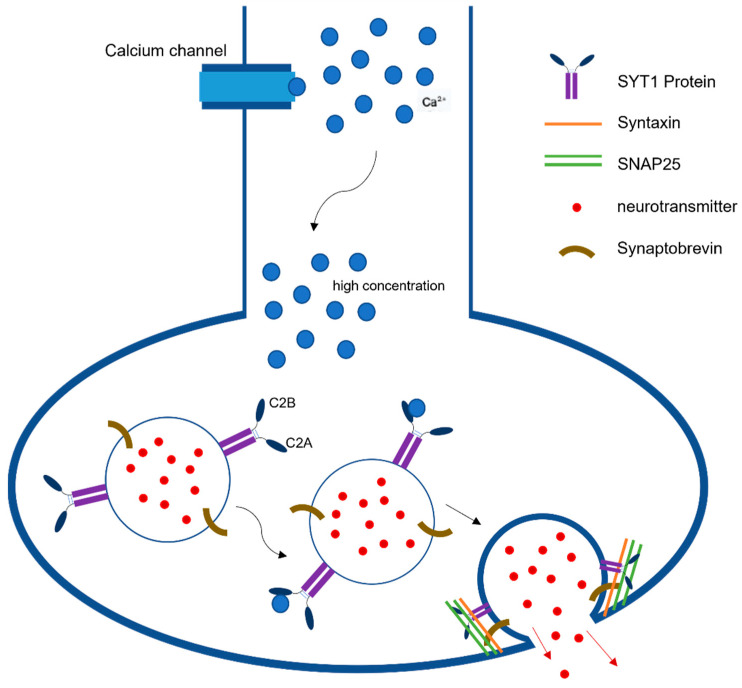
Synaptotagmin-1 Function in Neurotransmission. SYT1 plays a pivotal role in synaptic neurotransmission as a dual Ca^2+^sensor able to carry out both endocytosis and exocytosis independently. This figure describes the release of neurotransmitters following the detection of high concentrations of Ca^2+^ by SYT1’s C2B domains.

**Table 1 children-09-01439-t001:** Known Synaptotagmin Isoforms, Dysfunctions, and Clinical Mutations. Known synaptotagmin isoforms of clinical and functional relevance related to SYT1-associated neurodevelopmental disorder, including their noted neurotransmitter mechanism of release and related molecular dysfunction.

Isoform	Location	NT Mechanism of Release	Molecular Dysfunction	Clinical Symptoms of Mutation
**SYT1**	Synaptic vesicles and secretory granules of neuroendocrine of the hippocampus and forebrain	Independent fast synchronous NT release, clamping mini spontaneous release	Decreased evoked transmission	Developmental delay, sleep disturbances, EEG abnormalities, abnormal motor function, abnormal eye physiology
**SYT2**	Inhibitory neurons of the brainstem, spinal cord, cerebellum, and striatal neurons, neuromuscular junction	Evoked fast synchronous NT release	Impaired NT vesicle release	Moderate to severe motor deficits, muscle weakness, bulbar deficits, delayed motor development
**SYT7**	Hippocampus	SV priming and clamping, slow asynchronous NT release	Decrease RRP when KO alongside SYT1,	Mania, behavioral disturbances, bipolar-like behavior (in mice)
**SYT9**	Dense core vesicles and synaptic vesicles in limbic system and striatum, cortical and hippocampal neurons	Evoked fast synchronous NT release	Decrease in spontaneous miniature rate of synaptic vesicle fusion, inability to initiate membrane fusion	Not Available

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
