# Peer review of "SYT1-Associated Neurodevelopmental Disorder: A Narrative Review"

_children, 2022, doi:10.3390/children9101439_

Round 1

Reviewer 1 Report

Riggs and colleagues describe SYT1-related neurodevelopmental disorder and the functional implications. While well written, I'm not sure how much new information is present (or information that hasn't been published together before).

Firstly, there is a desperate need for figures. I would suggest a figure of the SYT gene family and one with the variants and consequences of those variants. While a relatively straightforward mechanism, it would benefit from an image to describe it. While present in Baker et al. and others, I think a table of the phenotypes would be valuable.

A bigger picture question I had was: Do you think there is a family of disorders for the SYT proteins? Sounds like SYT1 and SYT2 variants have similar consequences. It might be interesting to go through the other SYT proteins (maybe in a table) and discuss if they have clinical findings.

Some small things:

Section 1. Introduction

Line 26-27: use "autism spectrum disorder", not "autism spectrum disorders".

ASD is a neurodevelopmental disorder, not really a category of neurodevelopmental disabilities, remove this part of the sentence.

If you change "disorders" to "disorder" it changes have to has.

Line 29: schizophrenia is not a proper noun, lowercase "s"

Line 35: I would use developmental delay/intellectual disability OR cognitive impairment instead of intellectual development impairment.

Line 37: should read, “…composed of an N-terminal…”

Section 2. Synaptotagmin Isoforms and Function

Overall, I think you should use genes instead of isoforms, as this is a large gene family not different reading frames of one gene.

Ca2+ should be Ca2+

Line 63: “NT” is not previously defined.

Line 72: “SV” isn’t previously defined.

Line 102: there is an extra period at the end.

3. Synaptotagmins in Exocytosis and Endocytosis

Line 164: SYT1 is referred to as STY1.

Line 173: SYyntaxin-1 should be Syntaxin-1 I think.

Line 188-189: two typos with SYT1-associated disorder (one is just an extra space which could just be a formatting thing).

Line 202: since there are multiple transcripts, you should reference which one you are referring to for this variants.

Line 215: now you refer to it as “SYT1 Associated Neurodevelopmental Disorder.” Just needs to be consistent.

Were larger studies (like DDD, SSC, etc) queried for SYT1 variants? Either here or in previous studies?  It looks like there may be some relevant variants in denovo-db: https://denovo-db.gs.washington.edu/denovo-db/QueryVariantServlet?searchBy=Gene&target=syt1

Line 260: “mildly” should be “mild”

Line 261: you should refer to the patient not the variant (or both. Could say “…it was reported that the individual with the M303K variant…”

Line 273: what do you mean by “object mouth?” 

Overall:

It seems like PMID: 32362337 should be cited.

Riggs et al provide a nice summary of SYT1-related neurodevelopmental disorder, but I think there is some work necessary for it to be particularly useful as a resource for those studying or working with this disorder. Figures/Tables and further querying for variants would be necessary for publication.

Author Response

Please see attached responses for reviewer #1

Reviewer 2 Report

The manuscript titled "SYT1-associated neurodevelopmental disorder: a scientific and clinical review" reviews the literature relative to the current knowledge around the roles of the SYT family. These proteins are involved in sensing intracellular calcium and the docking of neurotransmitter vesicles to the membrane fusion complexes. Intuitively, disruption in their function has consequences on neurological functions. The authors are specifically relating SYT mutations to the symptoms associated with a recently identified disorder.

The manuscript is clear and well organised, however it lacks lack figures and probably depth. The addition of schematics illustrating the mechanisms and proteins involved in neurotransmitter release would certainly help the reader as well as a representation of the structure of SYT proteins . Grouping the symptoms of the disorder of interest into a table and signalling occurrence rate would not only make it easier for the reader but also an easily accessible resource for anyone interested in the disease rather than combing through the entire manuscript. The authors did not mention the age of diagnosis which would be of interest. Particularly since they mention severe motor issues in adolescent mice carrying a SYT2 mutation. Is the expression of SYT age-dependent or was this study carried out on adolescent mice and information for other developmental stages is not available? 

Minor comments:

Throughout the manuscript the authors use "NT" which has not been defined. I suspect it is just shorthand for "neurotransmitter" rather than a legitimate acronym. I recommend spelling out the word in full as the accumulation of acronym makes the text more tedious to read. 

Several typos where SYT becomes STY (line 139 and 169).

line 161: the date is missing on the Chapman citation

line 173: remove the first "Y" on syntaxin

line 195: replace "effects" by "affects", 2 different verbs, 2 different meanings

line 260: replace "mildly" by "mild"

Agree on how to refer SYT1 Associated Neurodevelopmental Disorder as over line 188, 189 and 216 it's referred to in  3 different way. 

Overall, this is a solid manuscript which could be published in the current form but could be improved on, particularly with the addition of illustrations.

Author Response

Please see attached responses for reviewer #2

Reviewer 3 Report

First of all, I would like to thank the authors in relation of the work they have done. 

The following are my comments that I believe should be resolved before the work can be considered for future publication: 

* Title: Add "Narrative" review. 

* Methods: add information about research strategy (if you used only Medline, how you selected the papers, the Mesh terms used...). Even if you are not performing a systematic review this information could be very interesting to emphasize the quality of the work. A flowchart of research strategy could be added at supplementary material.

* Introduction section: reorganize the content. First explain, the concept of synaptic function before trying to address the disorders in which the synaptic dysfunction have been described. Try to separate mutations related disorders with those with down or upregulation of expression of synaptic proteins. 

Consider the option to add a figure to summarize synaptic function and introducing the main topic of the review. 

A supplementary table about Synaptotagmins (SYTs) also could be of interest and could help to reduce the text. 

* Section Synaptotagmin Isoforms and Function: I recommend trying to summarize the content. Sentences that do not provide added information and text that is tedious to read. Value tables/figures to make it more visual.

* Combine Synatopgmin Function with the section Exocytosis and Endocytosis. It has more sense to divide the text between 2 sections of isoforms and other about function including exo and endocytosis. Evaluate the option to talk about (mention) the diseases that specifically have dysfunction of synatopgmins than can induce alteration of exocytosis and or endocytosis. 

* Attempt to establish a clearer genotype-phenotype relationship. Not as two separate watertight compartments/sections, because it is of less interest to the clinician. 

A panel with genotype-phenotype correlation can be of great interest.

* The conclusions are too long. It has to be a key idea. I think some of what is pointed out in this section could go to discussion. 

* Perhaps evaluate adding a paragraph or sub-section on future lines of research,.... specifically could be of interest. 

I would like to know the response to these comments before proceeding with a thorough review of the rest of the manuscript.

Author Response

Please see attached responses for reviewer #3

Round 2

Reviewer 1 Report

Riggs and colleagues addressed most of my concerns with their paper reviewing SYT1-associated neurodevelopmental disorder. I still have concerns that this is not a novel paper, and the lack of figures and tables makes it difficult to use as a review.

Specifically, a figure and table with variant and phenotypic information would be very useful. Having each patient in the text seems like a lot of extra text that could just be a table.

A few small things:

Abstract, line 10 should read "...resulting in a wide range of brain and neurodevelopmental disorders THAT are caused by mutations..."

Page 2 line 55 should read "THE mammalian synaptotagmin..."

Line 77: why are you defining SYT1 here? It should be in the earlier paragraph, if at all.

Page 4 line 143-144 is a run-on sentence.

Table 1 is missing headings.

Page 5 line 183 should read "THE C2B domain..."

This manuscript is a nice review of a syndrome that doesnt yet have a lot of information published. With some figures and tables, it will be a nice entry point for those studying SYT1-associated NDD.

Author Response

Please see attached response to reviewer #1

Reviewer 3 Report

I thank the authors for having made the suggested modifications. I believe that thanks to them the manuscript has been substantially improved. 

At present I would only make one last suggestion/recommendation which would be to move the section on search strategy to behind the introduction and put the heading methodology. 

Author Response

Please see attached response to reviewer #3
